# Structural Insights into the Dimeric Form of *Bacillus subtilis* RNase Y Using NMR and AlphaFold

**DOI:** 10.3390/biom12121798

**Published:** 2022-12-01

**Authors:** Nelly Morellet, Pierre Hardouin, Nadine Assrir, Carine van Heijenoort, Béatrice Golinelli-Pimpaneau

**Affiliations:** 1Institut de Chimie Des Substances Naturelles, Université Paris-Saclay, CNRS, UPR 2301, 91198 Gif-Sur-Yvette, France; 2Laboratoire de Chimie des Processus Biologiques, UMR 8229 CNRS, Collège de France Université, 75231 Paris, France

**Keywords:** RNase Y, ribonuclease, NMR, AlphaFold, coiled-coil, structure, dimerization

## Abstract

RNase Y is a crucial component of genetic translation, acting as the key enzyme initiating mRNA decay in many Gram-positive bacteria. The N-terminal domain of *Bacillus subtilis* RNase Y (Nter-BsRNaseY) is thought to interact with various protein partners within a degradosome complex. Bioinformatics and biophysical analysis have previously shown that Nter-BsRNaseY, which is in equilibrium between a monomeric and a dimeric form, displays an elongated fold with a high content of α-helices. Using multidimensional heteronuclear NMR and AlphaFold models, here, we show that the Nter-BsRNaseY dimer is constituted of a long N-terminal parallel coiled-coil structure, linked by a turn to a C-terminal region composed of helices that display either a straight or bent conformation. The structural organization of the N-terminal domain is maintained within the AlphaFold model of the full-length RNase Y, with the turn allowing flexibility between the N- and C-terminal domains. The catalytic domain is globular, with two helices linking the KH and HD modules, followed by the C-terminal region. This latter region, with no function assigned up to now, is most likely involved in the dimerization of *B. subtilis* RNase Y together with the N-terminal coiled-coil structure.

## 1. Introduction

In bacteria, gene expression in response to different environmental conditions is controlled by the fluctuation of messenger RNA (mRNA) transcript abundance [1,2,3]. The rapid mRNA turnover rate in bacteria (average half-life of several minutes) relies on the synthesis and degradation rates of each mRNA. RNases, responsible for RNA degradation, can be either very specific for their RNA substrate or be involved in bulk RNA turnover. The major RNases involved in bulk RNA turnover generally form multi-protein complexes for efficient and regulated degradation, which are called RNA degradosomes [4]. While most RNA synthesis machinery is similar in Gram-negative and Gram-positive bacteria, the machinery for bulk RNA degradation is quite different. In *Escherichia coli,* the well-studied hydrolytic endoribonuclease RNase E plays a major role in mRNA degradation and RNA processing [5], whereas RNase Y is one of the key enzymes directing RNA metabolism in *Bacillus subtilis* [6] as well as in other low-GC Gram positive species [7], including many pathogens [8,9,10]. Moreover, RNase Y was shown to be necessary for the full virulence of several Gram-positive bacterial pathogens [8,11]. Studying the function and structure of RNase Y is not only crucial to understand RNA metabolism in the many eubacteria that, unlike *E. coli*, do not rely on RNase E-based strategies of RNA degradation/maturation, but also to consider whether RNase Y could be targeted for the development of new antimicrobial agents against Gram-positive bacteria in the future, since RNase Y orthologs do not exist in eukaryotic cells.

While RNase Y and RNase E share many functions [12], they display only low sequence similarity (Figure 1). *E. coli* RNase E (1061 amino acids; 118 kDa) is composed of two major domains: the N-terminal domain that bears the endoribonuclease activity [13,14,15] and the C-terminal scaffolding domain that organizes the RNA degradosome [4,16,17,18] (Figure 1A).

This latter domain is intrinsically unstructured and unlikely to be extensively folded within the degradosome complex, except for a few microdomains that are conserved among RNase E orthologs [20,21]. The C-terminal microdomain, called the membrane targeting sequence, forms an amphipathic helix that interacts with the membrane [22]. Other conserved microdomains, ranging from 20 to 70 residues in size, are binding sites for RNA substrates and other protein components of the degradosome [20], including enolase [23,24] and PNPase [25]. *B. subtilis* RNase Y (529 amino acids; 58.9 kDa) is much shorter than *E. coli* RNase E and its catalytic domain is located at the C-terminus [12]. A transmembrane (TM) region, anchoring the protein to the membrane, is followed by the N-terminal domain and the C-terminal domain, itself composed of the catalytic domain with KH [26] and HD modules [27] and a C-terminal region of unknown function (Figure 1B).

Interestingly, RNase E was recently shown to be able to replace RNase Y in *B. subtilis* in vivo [28]. Efficient complementation of the *B. subtilis* ∆*rny* strain required RNase E to be localized to the inner membrane, while truncation of the C-terminal domain corresponding to the degradosome scaffold had only a minor effect.

The three-dimensional (3D) structure of RNase Y remains unknown. *B. subtilis* RNase Y lacking the TM region has previously been purified and analyzed by sedimentation velocity analytical ultracentrifugation, which showed that it is divided into an aggregated form and a mixture of dimers and tetramers [29]. The analysis of the interaction of the isolated domains (TM, N-terminal, and C-terminal) with each other, using a bacterial two-hybrid system [30], revealed that only the TM and the N-terminal domains showed self-interactions [12], suggesting that they were major contributors to the oligomerization of RNase Y. Although some bioinformatic programs (PONDR-FIT and metaPrDOS) predicted that the N-terminal domain is intrinsically disordered, others, such as COILS [12] or Lupas’ algorithm [31] (Appendix A) indicated that it is organized mainly as α-helices, adopting a flexible coiled-coil-like structure. In Lupas’ coiled-coil prediction [31], the probability of the N-terminal domain to form a coiled-coil structure was higher than 80% for residues 30–103, whatever the parameters used, and higher than 50% or 90% for residues 104–153. In addition, the protein fold recognition servers PSIPRED, PHYRE2, and the consensus secondary structure prediction server NPS@ predicted that the N-terminal domain adopts a helix-type secondary structure over almost the entire sequence (Appendix A).

We recently produced the N-terminal domain as a stand-alone protein called Nter-BsRNaseY and characterized its secondary structure by circular dichroism (CD) spectroscopy, size exclusion chromatography coupled with multi-angle light scattering (SEC-MALS), and size exclusion chromatography coupled with small-angle X-ray scattering (SEC-SAXS) [32]. Monitoring the weight-averaged molar mass as a function of protein concentration showed that the dimeric form of Nter-BsRNaseY is in equilibrium with a monomeric form, with a dissociation constant K_d_ of 1.3 µM. Nter-BsRNaseY was bound as a monomer when complexed with Fab (fragment antigen binding), which suggested that dissociation of the dimer could occur upon binding a protein partner [32]. Moreover, the dimer was shown to display an elongated form and a high content of α-helices [32].

*B. subtilis* RNase Y has been proposed to participate, together with polynucleotide phosphorylase PNPase, helicase CshA, and the glycolytic enzymes 6-phosphofructokinase and enolase in a multiprotein complex [33,34,35,36,37,38], similar to the RNase E-based degradosome in *E. coli* (Figure 1). Whereas PNPase is an RNA 3′ exoribonuclease and helicases are known to play important roles in remodeling RNA molecules, the function of the glycolytic enzymes within the complex is more elusive. Enolase is one intracellular/surface moonlighting protein present in many species, including eukaryotes and prokaryotes [39]. Inside the cell, it catalyzes the conversion of 2-phosphoglycerate to phosphoenolpyruvate in glycolysis. Yet, in some species, enolase is displayed on the cell surface, which allows it to play a role in bacterium–host interactions [40] by binding to host proteins such as plasminogen [41] and fibronectin [42]. It has been suggested that the N-terminal domain of RNase Y is the key element for assembling a degradosome complex in *B. subtilis* [33], similar to the C-terminal domain of RNase E in *E. coli* [41]. Therefore, as a first step to understand the functional importance of the N-terminal domain, here, we studied the structure of Nter-BsRNaseY using multidimensional heteronuclear NMR and the structure prediction algorithm AlphaFold [43]. Moreover, we also discuss the model of the full-length protein predicted by AlphaFold.

## 2. Materials and Methods

### 2.1. Prediction of the Secondary Structure of Nter-BsRNaseY

The secondary structure of Nter-BsRNaseY was analyzed with the protein fold recognition servers PHYRE2 (http://www.sbg.bio.ic.ac.uk/phyre2/html/page.cgi?id=index, accessed on 22 September 2022) [44], PSIPRED (https://bio.tools/psipred, accessed on 22 September 2022) [45], and the consensus secondary structure prediction server NPS@ (https://npsa-prabi.ibcp.fr/cgi-bin/npsa_automat.pl?page=/NPSA/npsa_seccons.html, accessed on 22 September 2022) [46]. The coiled-coil structure was predicted using (https://npsa-prabi.ibcp.fr/cgi-bin/npsa_automat.pl?page=/NPSA/npsa_lupas.html, accessed on 22 September 2022) either with or without a weight of 2.5 for positions ‘a’ and ‘d’ of the heptad repeat for windows of 14, 21, and 28 residues, knowing that the use of a weight for certain residues increases the prediction for a dimerization interface centered on these residues. The helical wheel plots were drawn with DrawCoil 1.0 [47].

### 2.2. Three-Dimensional Structure Prediction of Nter-BsRNaseY and Full-Length RNase Y Using AlphaFold

The AlphaFold models of Nter-BsRNaseY and full-length RNase Y were calculated for the untagged proteins with the Google Colab platform and AlphaFold2_advanced option https://colab.research.google.com/github/sokrypton/ColabFold/blob/main/beta/AlphaFold2_advanced.ipynb#scrollTo=ITcPnLkLuDDE, accessed on 22 September 2022 [43,48] that does not use templates (homologous structures) and refined using the Amber-relax option to enhance the accuracy of the side chains’ geometry. The default mode of sampling options was used: num_models = 5, ptm option, num_ensemble = 1, max_cycles = 3, tol = 0, num_samples = 1. The models were ranked according to their predicted local-distance difference test (pLDDT) confidence values (between 0 and 100, from low to high confidence). In the AlphaFold model file, the B-factor column for each residue is populated with its pLDDT value. For dimer prediction, a 1:1 value was input as the homo-oligomer assembly option. The structure figures were drawn with PYMOL [49].

### 2.3. Production of Nter-BsRNaseY

Methods for cloning the coding sequence corresponding to Nter-BsRNaseY (amino acids V24 to N192 of *B. subtilis* RNase Y), appended with a C-terminal hexahistidine tag, were described previously, as well as the production and purification of the protein [32]. Protein elution was followed by the optical density at 230 nm because of the absence of aromatic residues in the Nter-BsRNaseY sequence.

### 2.4. Culture and Purification of ^15^N-Labeled and ^13^C-^15^N-^2^D-Labeled Nter-BsRNaseY

For preparation of ^15^N-labeled Nter-BsRNaseY, cell growth was performed in 1L M9 minimal medium containing ^15^N-labeled NH_4_Cl, supplemented with 2 mM Mg_2_SO_4_, 4 g/L glucose, 10 µM CaCl_2_, 1 mg/L biotin, 5 mg/L thiamine, 50 µg/mL ampicillin, and 30 µg/mL chloramphenicol. For preparation of ^13^C-^15^N-^2^D-labeled Nter-BsRNaseY, cell growth was performed in 1L of M9 minimal medium containing D_2_O and ^15^N-labeled NH_4_Cl, supplemented with 2 mM MgSO_4_, 3 g/L [^13^C]-glucose, 10 µM CaCl_2_, 1 mg/L biotin, 5 mg/L thiamine, 50 µg/mL ampicillin, and 30 µg/mL chloramphenicol. The proteins were purified by Ni-NTA affinity and SEC, as described [32], then by SEC on a Superdex Increase 200 10/300 GL increase column in 20 mM HEPES pH 7.5, 500 mM NaCl, 10% glycerol for ^15^N-labeled Nter-BsRNaseY, or 40 mM MES pH 6.8, 200 mM NaCl for ^13^C-^15^N-^2^D-labeled Nter-BsRNaseY (Appendix A). Finally, ^15^N-labeled and ^13^C-^15^N-^2^D-labeled Nter-BsRNaseY were concentrated to 10 mg/mL (240 µM) or 40 mg/mL, respectively, using Amicon concentrators (30 kDa cutoff, Milllipore), then aliquoted, frozen in liquid nitrogen, and stored at −80 °C. 

### 2.5. Circular Dichroism of Nter-BsRNaseY

The far-UV CD spectrum (195−260 nm) of Nter-BsRNaseY was recorded at various temperatures (293, 298, 300, 303, 313, and 318K) on a Chirascan-plus CD spectrometer (Applied Phtophysics, Surrey, UK) (Appendix A). Spectra of Nter-BsRNaseY (450 µM) were acquired in quartz cuvettes of 0.01 mm optical path length in 40 mM MES pH 6.8, 200 mM NaCl. A resolution of 1 nm, bandwidth of 1 mm, and time per points of 1s were applied. All spectra, resulting from an average of ten accumulations, were corrected from buffer contribution.

### 2.6. NMR Resonance Assignments for RNase Y Backbone 

NMR samples of ^13^C-^15^N-^2^D-labeled Nter-BsRNAseY (970 µM) were prepared in 40 mM MES buffer pH 6.8, 200 mM NaCl, 5% D_2_O for the reference condition. A series of 2D ^1^H-^15^N BEST-Transverse relaxation optimized spectroscopy (TROSY) correlation spectra were recorded at 278K, 288K, 293K, 298K, 303K, 308K, 313K, and 318K, and a series of 3D BEST-TROSY (HNCA, HNCOCA, HNCACB, HNCOCACB, HNCO, and HNCACO) correlation spectra were collected at 300, 303, and 313K on a Bruker AVANCE III HD 950 MHz equipped with a TCI cryoprobe. A series of BEST-TROSY correlation spectra were also recorded at 303K at various concentrations of Nter-BsRNAseY (50, 98, 185, and 332 μM). A 3D ^1^H-^15^N nuclear Overhauser effect spectroscopy (NOESY)-heteronuclear single quantum correlation (HSQC) spectrum, with mixing times of 200 ms, was also collected to help peaks assignment at 303K. Data processing and analysis were performed using the Topspin^®^ 4.0 and CcpNmr version-2 software [50]. To record the spectrum of the completely denatured protein, the buffer was supplemented by 6 M urea. The ^15^N R_1_ and R_2_ relaxation rates and {^1^H}-^15^N heteronuclear nuclear Overhauser effects (NOEs) were measured at 303K. The ^15^N R_1_ and R_2_ relaxation experiments were based on the refocused ^1^H-^15^N HSQC relaxation experiments and recorded in an interleaved pseudo-3D method with an inter-scan delay of 4 s. For the determination of R_1_ relaxation rate constants, 11 total data sets were collected at relaxation delay times of 10, 50, 100, 200, 400, 500, 600, 800, 1000, 1500, and 2000 ms. For the determination of R_2_ rate constants, 12 data sets were collected at delay times of 17, 34, 51, 68, 84.8, 102, 136, 170, 204, 237, 271, and 305 ms. R_1_ and R_2_ spectra were recorded as 144 × 2024 complex data points. For the backbone {^1^H}-^15^N heteronuclear NOEs, two different spectra were recorded as 256 × 2048 complex data points in an interleaved manner with and without a 4 s proton saturation pulse. The R_1_ and R_2_ rates, heteronuclear NOE values, and their associated errors were determined from the peak intensities using the CcpNmr version-2 software [50].

The amide proton chemical shift temperature coefficients (∆δHN/∆T (ppb/K)) were calculated using the Shift-T web server (http://biophysical.science/shiftt, accessed on 22 September 2022) [51]. 

## 3. Results

### 3.1. Several Regions of Nter-BsRNaseY Have a High Propensity to Form α-Helices

To study the structure of Nter-BsRNaseY by NMR, the protein was expressed in *E. coli* and labeled with ^15^N or ^15^N, ^13^C, and ^2^D. Its ^1^H-^15^N BEST-TROSY spectrum was first recorded at 300K and a concentration of 970 µM, under which conditions Nter-BsRNaseY is a dimer [32]. The TROSY spectrum, which provides correlations between nitrogen atoms and amide protons, shows a narrow chemical shift dispersion in the ^1^H dimension (gathered in the 7.9–8.5 ppm region) that is characteristic of rather disordered or α-helices-forming residues (Figure 2A,B). 

To confirm that the protein contains ordered structural elements, the ^1^H-^15^N heteronuclear multiple quantum coherence (HMQC) spectrum was compared to that of a denatured sample (Appendix A). The addition of 6 M urea, a widely used protein denaturing agent [52], resulted in substantial sharpening of the chemical shifts for most signals, as well as in several significant chemical shift variations. These changes indicate the disappearance of structured elements, probably resulting from a change of inter-molecular interactions.

To obtain more information about the secondary structure of the different regions of Nter-BsRNaseY at the residue level, we assigned the amide ^1^H, ^15^N and ^13^C resonances using the standard triple experiments at 300, 303, and 313K (Appendix A). The CD spectrum of BsRNAseY was also recorded at these temperatures, as well as at 298 and 318K, to monitor the signal at 220 nm and thus the variation in the α-helices content with temperature (Appendix A). We assigned the resonances for 157 out of 176 (89%) of the Nter-BsRNaseY backbone atoms. Assignment is missing for the three N-terminal amino acids (M23-R25) and for residues H67-K68, R81, H102, D116-S118, R122-H125, M139-Q140, M161-R162, and H169. We note that the five histidine residues of Nter-BsRNaseY (H67, H102, H117, H125, and H169) are localized within these segments. The Nter-BsRNaseY ^13^CO, ^13^C_α_, ^13^C_β_, ^1^H_N_, and ^15^N chemical shifts were then used as input for the TALOS-N software [53] to predict the protein backbone torsion angles along the sequence. The TALOS-N results reveal that several regions of the protein (residues 39–61, 73–88, 122–157, and 171–191) have the highest propensity to form α-helices at 300K (Figure 3A and Appendix A). 

In regular turns and α-helical polypeptide chains, sequential amide protons close in space (HN_i_-HN_i+1_) yield to cross-peaks of high intensity in the NOESY spectra, while protons farther away (HN_i_-HN_i+2_) give peaks of lower intensity. We analyzed the ^1^H-^15^N (HSQC)-NOESY 3D spectrum to delineate the helical motifs (Appendix A). Despite a lot of overlapped peaks, we were able to find NOEs corresponding to several amide protons HN32-HN34, HN55-HN57, HN109-HN112, HN136-HN138, and HN142-HN144 (Appendix A). HN55-HN57 is located in the first helix predicted by TALOS-N, HN109-HN112 is located in the second helix, whereas HN136-HN138 and HN142-HN144 are located in the third helix (Appendix A).

The 133–149 helix appears to be the most stable helix since it is present at both 300 and 313K (Figure 3A), with the same predictions for all amino acids. Interestingly, we observed a significant increase in the percentage of α-helices as the temperature decreases (36% at 313K but 65% at 300K), as predicted by TALOS-N (Appendix A). This increased folding of the protein with a decrease in temperature, with a break in the α-helix content between 303 and 300K, was also observed on the CD (Appendix A) and ^1^H spectra (Appendix A). Indeed, a decrease from 318K to 288K, leads to a deshielding (shifting to higher ppms) of some amide protons (between 8.4 and 8.6 ppm), a higher chemical shift dispersion of peaks between 6.9 and 7.7 ppm (Figure 3B and Appendix A), as well as a shielding of several methyl groups around 0.4 ppm (Figure 3C and Appendix A). Numerous amide proton temperature coefficients (∆δHN/∆T) were higher than −5 ppb/K (Figure 3D), indicating a high probability for these amide protons to be hydrogen bonded [54]. Interestingly, these residues belong mainly to the regions of the protein described above that have a high propensity to form α-helices.

To obtain further insights into the flexibility of Nter-BsRNaseY, we analyzed the changes in the ^1^H-^15^N TROSY cross-peak intensities as a function of temperature (Figure 3E). In such spectra, the intensity of the peaks varies with the mobility of the corresponding residues, with the higher flexible regions showing higher peak intensities. We observed that the N-terminal residues (20 to 35) and the C-terminal extremity (residues 176–192) of Nter-BsRNaseY belong to the most flexible regions of the protein. For the N-terminal region, the peak intensities showed little variation with temperature, indicating that flexibility was inherent to this region and not dependent on temperature. In contrast, whereas the C-terminal region was only slightly more flexible than the rest of the protein at low temperature (278K), its flexibility increased much faster with temperature than the rest of the protein (Figure 3E). 

### 3.2. Two Main Conformations of the C-Terminal Extremity of Nter-BsRNaseY

The analysis of the ^1^H-^15^N BEST-TROSY spectra of Nter-BsRNaseY at various temperatures (Appendix A) allowed us to assign two sets of amide peaks corresponding to two conformations for several residues belonging to the C-terminal extremity (residues 171–191 (Figure 2C and Appendix A). Yet, it was not possible to assign duplicated peaks for residues N180, R181, E184, and E185 because their intensity was too low to be detected in the 3D spectrum (Appendix A).

The relative intensity of the duplicated cross-peaks decreased with temperature, indicating a change in conformers ratio (Appendix A). Between 293 and 308K, the duplicated cross-peaks were clearly visible, suggesting that residues 171–191 switch between two conformations at a slow rate on the NMR time scale (~millisecond range). Above 308K, only one cross-peak per residue was observed, indicating the presence of a single conformer. Interestingly, we noticed that the intensity ratio between the pairs of peaks of residues 173–176 is lower than that of residues 182–191 (Figure 3F). This indicates that the C-terminal residues (182–191) are more sensitive to temperature than residues 173–176. 

In addition, we recorded a series of BEST-TROSY spectra at various concentrations of Nter-BsRNAseY (50–332 µM) and 303K in order to evaluate the influence of protein concentration on the duplicated cross peaks in the ^1^H-^15^N BEST-TROSY spectra (Appendix A). At all these concentrations, Nter-BsRNaseY was shown to be a dimer [32]. The comparison of the spectra of Nter-BsRNaseY at low and high concentrations (50 μM and 970 μM, respectively) showed no significant chemical shift variations. However, peaks duplication was not observed at low protein concentration for the C-terminal residues (residues 171–191) (Appendix A). This suggests a concentration-dependent change in the equilibrium between two species that display different conformations of the C-terminal residues.

### 3.3. High Flexibility of the N- and C-Terminal Residues

In addition, we also observed variations in the peaks in the ^1^H-^15^N BEST-TROSY spectrum, with some of them showing a significantly weaker intensity than the others (Figure 2B). To determine if this results from a conformational change of the protein, we studied the dynamics of the Nter-BsRNaseY backbone using NMR ^15^N relaxation, which is a powerful tool to characterize dynamic processes of proteins in solution over a wide range of time scales [55]. Indeed, on one hand, fast motions (picosecond to nanosecond scale) can be characterized by heteronuclear ^15^N longitudinal relaxation rate (R_1_), transverse relaxation rate (R_2_), and ^15^N-{^1^H} heteronuclear NOE (hetNOE) of amide group resonances; on the other hand, chemical exchange mechanisms are generally involved in movements on the microsecond-millisecond scale and contribute to the R_2_ transverse relaxation rate. Thus, heteronuclear NOEs are very sensitive to local mobility, with large NOE values indicating restricted motion. 

NMR ^15^N relaxation measurements at 950 MHz ^1^H and 303K of Nter-BsRNaseY show that the R_1_ and hetNOE relaxation rates are relatively homogeneous over the regions encompassing residues 40 to 115 and 170 to 191 (Figure 4A,C). 

Higher R_2_ and hetNOEs values than average were observed for residues belonging to the 127–155 segment (Figure 4B,C), suggesting reduced mobility. Accordingly, up to 313K, this region was shown to have a high propensity to adopt an α-helix fold (Figure 3A). On the contrary, the N-terminal extremity (residues 24–39) showed lower R_2_ and hetNOE values than the rest of the polypeptide chain, indicating a high flexibility of this segment. Moreover, for the segment 158–170, low hetNOE values were observed (Figure 4C), indicating an increase in mobility, as well as high ^15^N R_2_ values (Figure 4B), likely resulting from a contribution to µs-ms conformational or chemical exchange. These observations are consistent with the α-helix predictions (Figure 3A), which show that this segment has a low propensity to form an α-helix. 

### 3.4. AlphaFold Models of Nter-BsRNaseY 

To calculate 3D models of Nter-BsRNaseY, we used the recently released AlphaFold algorithm [43], which has revolutionized structural biology by its highly accurate predictions of protein structures (Figure 5). 

AlphaFold was used to predict the 3D structure of both the monomer (Figure 5A) and the dimer (Figure 5B,C) because previous studies indicated that, at the concentration of Nter-BsRNaseY used for the NMR studies, Nter-BsRNaseY is in the dimeric form [32]. For both the monomer and dimer models, the AlphaFold prediction is highly reliable for residues 38 to 169, with predicted pLDDT values over 90 (Figure 5D–F), which means that, in addition to a good backbone prediction, the side chains are also correctly oriented. Nevertheless, the pLDDT values decreased gradually after residue 170—but still remained above 80 until residue 188—for the best model of the monomer (Figure 5D) and the dimer (Figure 5E,F), indicating that the position of the C-terminal residues is less well predicted.

The monomer is constituted of a long α-helix (residues 23 to 149), followed by a turn (residues 150 to 151) and a C-terminal extremity (residues 152–192), which forms either a long straight helix or a curved helix with a kink around residues 166 to 169 (Figure 5A). Interestingly, in all dimer models, the interaction between the two monomers involves a long parallel coiled-coil structure containing two long α-helices (residues 23 to 149) (Figure 5B,C). No antiparallel coiled-coil structure was found in the best predictions for the Nter-BsRNaseY dimer by AlphaFold. In addition, the two long α-helices wrap around each other to form a left-handed supercoiled structure, but with a break in the supercoil around residues 62–72 (Figure 5C). The C-terminal extremity of the dimer is formed by a pair of helices, one straight helix from one monomer and one curved helix from the second monomer. 

In a perfect coiled-coil structure, two α-helices pack against each other, with the amino acid side chains adopting a well-established architecture that is called a knob-into-hole (KIH) packing [56,57]. The typical coiled-coil sequence consists of a series of adjacent heptad repeats (a-b-c-d-e-f-g)_n_, in which the residues at position ‘a’ are either apolar or charged (R, K, E and D) and the residues at position ‘d’ are usually hydrophobic (V, L, I). The long α-helix of the N-terminal part (residues 30–149) of Nter-BsRNaseY satisfies the conditions for forming a coiled-coil structure in a parallel orientation, involving extensive ionic and hydrophobic interactions, as illustrated by the helical wheel plots (Appendix A). 

Hydrophobic side chains are present at the ‘a’ and ‘d’ positions of the 30–71 segment (Appendix A), leading mainly to hydrophobic interactions between the two chains (Appendix A), whereas hydrophobic side chains are present only at the ‘d’ positions of the 74–122 and 109–150 segments (Appendix A), leading to both hydrophobic and electrostatic interactions (Appendix A). In addition electrostatic interactions are observed between residues located at the ‘e’ and ‘g,’ positions (Appendix A), ‘d’ and ‘e’ positions (Appendix A), or ‘a’ and ‘g’ positions (Appendix A). The charges of the residues located inside the helix, at positions ‘a’, ‘d’, ‘e’, and ’g’ of the heptad repeat are of opposite sign, forming ionic interactions that stabilize the structure (yellow circles in Appendix A). 

However, the N-terminal region (residues 30–71) does not adopt a canonical coiled-coil structure (Figure 5C), with many alanine residues being present at the ‘a’ position (instead of R, K, E and D) or ‘d’ position (instead of V, L, I), together with leucine and isoleucine (Appendix A). The break in the supercoil noticed around residues 62–72 could result from the differences in the dimerization interfaces of the segments containing residues 30–71 (Appendix A) on one hand and those containing residues 74–122 (Appendix A) on the other hand. 

Finally, the intermolecular interactions between the helical C-terminal extremities (residues 152–192) of each monomer involve hydrophobic and electrostatic interactions, but do not have the characteristics of a coiled-coil structure (Appendix A). The kink in one helix (Figure 5B,C) is maintained through hydrophobic interactions between the two chains involving residues I158, I159, L160, V163, L167, I171, and M174 from each monomer (Figure 5G and Appendix A). Therefore, the different conformations of the two C-terminal extremities of the Nter-BsRNaseY dimer come from a different structural environment of residues 158–192 in the two chains. 

### 3.5. AlphaFold Model of the Full-Length B. subtilis RNase Y

Finally, we also used AlphaFold to model the structure of the monomer and dimer of full-length *B. subtilis* RNase Y (Figure 6 and Appendix A) and fine-tune the boundaries of the C-terminal domain (Figure 1B) based on structural elements. 

As for Nter-BsRNaseY, in the monomer of *B. subtilis* RNase Y, the N-terminal domain is formed by a long α-helix composed of residues 23–147, followed by a turn involving residues 150–151 and a region, composed of residues 155–202, that adopted either a long straight helix or a curved helix with a kink around residues 167–171 (Figure 6A). Hence, some residues (193–202, Figure 1B), initially thought to link the N-terminal domain and the C-terminal globular domain, appear to extend the long α-helix (Appendix A). The superposition of the five best AlphaFold models of the full-length *B. subtilis* RNase Y monomer reveals a high mobility of the N-terminal and C-terminal domains relative to each other, with a hinge around residues 149-152 belonging to the turn (Figure 6A). This flexibility suggests that the turn in the coiled-coil structure is functionally relevant.

The fold of the globular C-terminal domain (residues 211–520), which includes the catalytic domain and a C-terminal region with unknown function (Figure 1B), consists of numerous α-helices and a few β-sheets (Appendix A). The RNA binding module KH (residues 211–276), composed of three short α-helices and a three-strands β-sheet, is linked to the HD module (residues 336–429), composed of five α-helices, by two helices (residues 281–328). These three structural elements pack together to form the globular catalytic domain (residues 211–429), which is linked to the C-terminal region of unknown function (residues 438–520), composed of two α-helixes and a three-strands β-sheet, via a flexible linker. 

The model of the full-length *B. subtilis* RNase Y dimer (Figure 6B–D) shows that dimerization involves both the N-terminal coiled-coil and the C-terminal domain. The supercoiling of the long N-terminal α-helices is well conserved between the models of the dimers of Nter-BsRNaseY and the full-length protein. Interestingly, the ten last C-terminal residues of the C-terminal region with unknown function are disordered in the monomer (Appendix A) but completely structured, extending the last β-strand, in the dimer (Appendix A). This structuration allows the C-terminal region to participate to the dimer interface, through packing interactions between the long last β-strand from each monomer (residues 505–519) (Appendix A). 

In most models of the full-length *B. subtilis* RNase Y (rank 1 to 4 for the monomer and rank 1 to 3 for the dimer), the globular C-terminal domain acts as an extension of the last helix (residues 152–192) observed in the model of Nter-BsRNaseY (Figure 6). Yet, in the less likely structures (rank 5 for the monomer and ranks 4 and 5 for the dimer), the globular domain folds up on the long N-terminal coiled-coil (Figure 6A,B). It remains to be seen whether this fold, made possible by the flexibility of the N- and C-terminal domains relative to each other, is physiologically relevant or not.

The predictions of AlphaFold for the full-length RNase Y in its monomeric/dimeric forms are still of good quality, although slightly lower than those for Nter-BsRNaseY (Appendix A). Indeed, average pLDDT values were found to be 90.5/81.1 for the coiled-coil domain (residues 22–149) and 90.6/87.1 for the catalytic domain (residues 211–429), while residues with pLDDT values between 70 and 90 are expected to be modeled with an overall good backbone prediction [43]. Within the catalytic domain, the pLDDT values were 85.9/81.6 for the KH domain, 93.0/91.0 for the α-helices between the KH and HD domains, and 93.5/90.2 for the HD domain. The lowest pLDDT values were 84.7/77.4 for the C-terminal region of unknown function and 83.5/83.8 for residues 152–202 that correspond to the C-terminal extremity of the Nter-BsRNaseY construct, formed by one straight and one curved helix. Yet, the pLDDT value for this region was over 80 for the best model of the dimer, indicating that the predicted fold is still very likely. The loops connecting secondary structure elements (indicated by stars in Appendix A) had lower pLDDT values than average, as expected.

## 4. Discussion

Here we studied the N-terminal domain of *B. subtilis* RNase Y using NMR to investigate its dimerization mode. Indeed, at the concentration used for the NMR studies (50–980 µM), Nter-BsRNaseY was in the dimeric form [32].

### 4.1. Interpretation of the NMR Data of Nter-BsRNaseY Using AlphaFold

The 3D structures of the Nter-BsRNaseY monomer and dimer were modeled with AlphaFold and analyzed in light of the NMR data (Appendix A). The NMR data showed that Nter-BsRNaseY is mainly in the helical form (65% at 300K) and that the helices ratio increased with decreasing temperature, as shown by the 1D spectra recorded from 318K to 288K (Figure 3B and Appendix A). The AlphaFold model of the Nter-BsRNaseY dimer exhibited a long N-terminal parallel coiled-coil, followed by a turn, predicted to involve residues 149–151, and C-terminal helical ends that take part in dimer formation and can adopt two main conformations (Figure 5). Accordingly, it had previously been shown by analyzing the binary interactions of the isolated domains of *B. subtilis* RNase Y that the N-terminal domain was a major contributor to the oligomerization of RNase Y [12]. The AlphaFold models appear to be more representative of what would happen at lower temperatures than those used to record the NMR spectra because these models contain more helices (97%, with 78% of the residues having pLDDT values over 90 for the best AlphaFold model).

The Nter-BsRNaseY dimer was found to form a parallel, and not an antiparallel coiled-coil helix, in agreement with the nice fit of the parallel model to the SEC-SAXS experimental data [32]. This fold, which is favored by hydrophobic and electrostatic interactions, as shown by the helical wheel diagrams (Appendix A), allows the anchoring of the N-terminal end of Nter-BsRNaseY to the cytoplasmic membrane via a transmembrane region (Figure 1B). Accordingly, all models generated with AlphaFold for the full-length RNase Y dimer also involved long parallel coiled-coils involving residues 1–149 (Figure 6B). 

The α-helices predicted by TALOS to be the most temperature stable (residues 39–61, 73–88, and 122–149) correspond to a helical fold in the AlphaFold models (Appendix A). The most hydrophobic residues of these helices are involved in the dimerization surface in the best AlphaFold model (Appendix A). Moreover, the TALOS-N predictions from our NMR data show a decrease in the propensity of residues 62–72 to be structured into a continuous helix (Figure 3A, Appendix A) that agrees with the break in the supercoil present in the AlphaFold models around this position. The helices predicted from the NMR data have the same boundaries as those in the AlphaFold models, except for residues 89–121 (Appendix A). Indeed, this region is fully part of the helical dimeric interface generated by AlphaFold, whereas it is more flexible experimentally according to the NMR data obtained at 300K. This difference could be explained by the low ratio of hydrophobic residues in this region—i.e., present only at the position “d” of the heptad repeats (Appendix A)—and by the high number of electrostatic interactions stabilizing this segment in the AlphaFold model (Appendix A). The electrostatic interactions are probably reduced in the NMR experiment due to the addition of 200 mM NaCl in the samples, thus explaining the observed flexibility. 

According to the backbone chemical shifts and ^15^N relaxation experiments (Figure 3 and Figure 4), the N-terminal extremity of Nter-BsRNaseY was shown to be quite flexible. This flexibility agrees with the low pLDDT values for this region in the AlphaFold models (Figure 5D–F).

The ^15^N relaxation studies also indicated that the C-terminal extremity of Nter-BsRNaseY (residues 158–170) is rather flexible. This flexibility agrees with the AlphaFold models, in which it formed a dimer, with each monomer being constituted of either a bent or straight helix (Appendix A). Moreover, splitting of the ^1^H-^15^N HSQC signals was observed for several peaks belonging to the very C-terminal end (residues 171–191) (Figure 2C and Appendix A), which is rich in both positively and negatively charged residues and also contains a stretch of hydrophobic residues (Appendix A). This splitting may result from a slow exchange on the NMR chemical shift time scale between at least two different conformations involving these last 20 residues, as proposed in the AlphaFold model (Appendix A), where residues 158–192 of the two interacting chains have a different environment. The duplicated peaks disappeared upon an increase in temperature or a decrease in protein concentration, suggesting that dissociation of the dimer occurs at the very C-terminal end of Nter-BsRNaseY.

The relevance of the AlphaFold models was consolidated by our analysis of the HSQC-NOESY 3D spectra (Appendix A), since we were able to assign many intense HN_i_-HN_i+1_ and several low HN_i_-HN_i+2_ cross-peaks. Thus, altogether, the NMR data nicely support the α-helical fold of Nter-BsRNaseY proposed by AlphaFold, as well as its dimerization mode, not only at the N-terminal but also at the C-terminal end.

Finally, the best AlphaFold prediction for Nter-BsRNaseY was analyzed with DALI [58] to find the closest protein structures in the PDB, with a strong match being defined by a Z-score > (n/10) − 4, with n the number of residues. The highest structural homology was found for the coiled-coil domain of SAS-6, a centriole protein (PDB code 6YRN, 11% sequence identity, Z score = 7.9, root mean square deviation (rmsd) of 3.9 Å for 111 aligned Cαs) [59]. SAS-6 fragments were shown to be organized as two-stranded parallel coiled-coil domains that could form higher-order interactions: nine SAS-6 dimers associated via interactions between their N-terminal globular head domain to form a ring. In addition, asymmetric parallel association between coiled-coil domains of SAS-6 was found to be important to form a cartwheel structure and provide polarity to the assembly.

### 4.2. AlphaFold Model of the Full-Length B. subtilis RNase Y

To complement our NMR study, which was performed using a truncated *B. subtilis* RNase Y protein that lacks the C-terminal globular domain (residues 193–520), AlphaFold was used to generate a model of the full-length *B. subtilis* RNase Y, which was shown to exist as a dimer and possibly also as higher oligomeric forms [12]. Interestingly, in the model of the full-length protein, the coiled-coil structure observed for the Nter-BsRNAseY construct was extended by ten residues (Appendix A) that were previously thought to belong to the C-terminal domain. Analysis of the assemblies with the Proteins Interfaces Structures and Assemblies (PISA) program [60] indicates that the coiled-coil structure contributes to ~72% of the buried surface area in the full-length dimer. Moreover, the models unveiled the probable fold of the C-terminal domain, which was unknown until this point. Intriguingly, the pLDDT values, especially for region 130–190, were lower for the full-length RNAse Y (Appendix A) than for Nter-RNAseY (Figure 5D–F). This suggests an effect of the C-terminal domain on the conformation of residues 130–190.

The analysis of the best AlphaFold model of the C-terminal domain of *B. subtilis* RNase Y with DALI [58] revealed high structural similarity with proteins of known structure containing the HD or KH domains (Figure 1). Indeed, the highest structural homology was found for a protein predicted to belong to the HD hydrolase superfamily (PDB code 2PQ7, 24% sequence identity, unpublished), with a Z-score of 12.3 and an rmsd of 3.4 Å for 139 aligned Cαs. Interestingly, the latter protein was crystallized in the presence of divalent Fe atoms and superposition of the HD domains of both proteins highlights the potential binding mode of divalent metal ions to *B. subtilis* RNase Y, involving conserved His/Asp residues of the HD domain (Appendix A). Indeed, it is known that cleavage by RNase Y requires the presence of Mg^2+^ ions, which can be replaced by Mn^2+^ or Zn^2+^ [6]. In addition, the C-terminal domain of *B. subtilis* RNase Y showed a high structural homology with the putative RNA-binding protein of the exosome complex (PDB code 2Z0S, 21% sequence identity, unpublished), with a Z-score of 12.2 and an rmsd of 1.8 Å for 81 aligned Cαs, and with the KH-containing RNA-binding protein RRP4 of the archaeal exosome (PDB code 2BA0, 21% sequence identity), with a Z-score of 11.8 and an rmsd of 1.7 Å for 78 aligned Cαs [61]. The exosome is a large multi-subunit RNase complex that is required for 3′/5′ processing of ribosomal RNA in archaea and eukaryotes and thus it has a similar function to RNase Y. The exosome structure consists of six RNase phosphorolytic (PH) domain subunits forming a hexameric ring, which associates with three KH and/or S1-containing subunits to form a regulatory RNA recognition platform that restricts entry to the catalytic chamber to unstructured RNA substrates. 

The analysis of the C-terminal region with DALI identified several cation-binding proteins as close homologs, such as the cation efflux protein MamM (PDB code 3W64, 14% sequence identity, Z-score of 9.3, rmsd = 2.0 Å for 73 aligned Cαs) [62] and the cytoplasmic C-terminal domain of zinc transporter protein YiiP (PDB code 3H90, 11% sequence identity, Z-score of 7.8, rmsd = 2.3 Å for 72 aligned Cαs) [63]. MamM is one of the main ion transporters of magnetosomes, i.e., bacterial organelles that enable magnetotactic bacteria to orientate along geomagnetic fields. It was shown that the cytosolic domain of MamM forms a stable ‘V-shape’ dimer that undergoes distinct conformational changes upon divalent cation binding ([62], whereas the C-terminal domain of YiiP adopts a metallochaperone-like fold that allows it to deliver zinc ions to protein targets. 

Finally, the C-terminal region of *B. subtilis* RNase Y was found to be homologous with proteins that are components of megadalton-sized ring-shaped complexes, such as protein PRGK (PDB code 2Y9J, 16% sequence identity, Z-score of 7.2, rmsd = 2.0 Å for 69 aligned Cαs) [64], or the flagellar M-ring protein FliF (PDB code 6SD3, 9% sequence identity, Z-score of 6.9, rmsd = 2.5 Å for 70 aligned Cαs) [65]. The homology of the C-terminal region of RNase Y with the abovementioned proteins should prompt investigation of whether it could be involved in the formation of a high-molecular weight oligomer. Finally, the structural analysis showed that the C-terminal domain of RNase Y is not structurally related to the N-terminal catalytic domain of RNase E.

The AlphaFold models of the full-length dimer of *B. subtilis* RNase Y also revealed a potential function of the C-terminal region as a dimerization domain. This is supported by the analysis of the assemblies with the PISA program [60], which indicated that, in solution, the C-terminal region should contribute to ~10% to the buried surface area in the full-length dimer. The dimerization function of the C-terminal region was not expected since previous bacterial two hybrid systems experiments did not reveal self-interactions of this region [12]. Furthermore, the dimeric interaction that involves the C-terminal regions of two monomers generates a hole in the structure of the dimer close to the KH domain (Appendix A), suggesting that this hole, with a diameter of 12–13 Å, could be used to bind RNA. In this way, the KH domain could play a role in restricting access to the central chamber to unstructured RNA substrates, in a similar way to the RNA-binding module of the exosome complex [61]. This hypothesis agrees with the global electrostatic charge distribution of the dimeric C-terminal domain of *B. subtilis* RNase Y (Appendix A), in which the KH domains, the last C-terminal β-sheets, and the residues forming the hole are positively charged and prone to bind a negatively charged molecule, such as RNA. 

### 4.3. Interaction with Cellular Partners of the Degradosome Complex

*B. subtilis* RNase Y fulfills a similar function to *E. coli* RNase E, which itself participates in a degradosome complex (Figure 1). In *E. coli*, RNase E forms the core of the degradosome, which in its minimal version includes polynucleotide phosphorylase (PNPase), the ATP-dependent DEAD-box RhlB RNA helicase (RhlB), and enolase [66] (Figure 1A). Several studies have suggested that the carboxy-terminal domain of RNase E acts as a flexible tether of the degradosome components. Our study shows that the C-terminal domain of *B. subtilis* RNase Y is not structurally related to the N-terminal catalytic domain of RNase E (Figure 1) [28]. Nevertheless, intriguingly, it has been previously shown that *E. coli* RNase E can effectively replace RNase Y in *B. subtilis* [28], and that the presence of the RNase E degradosome scaffold was not crucially important for the capacity of RNase E to complement for RNase Y. 

However, similarly to *E. coli* RNase E, the N-terminal domain of *B. subtilis* RNase Y is flexible relative to the C-terminal globular domain, due to a turn involving residues 150–151, suggesting that the N-terminal domain could act as a tether to assemble the other degradosome proteins. In particular, several regions of the N-terminal region, such as peptide 62–72, which delineates a break in the supercoil, and peptide 89–121, which was shown to be the most flexible part in the coiled-coil, are likely candidates for binding degradosome proteins. Indeed, such interaction of a flexible protein region with several cellular partners (RNA or proteins) has previously been reported for other RNA-binding proteins [67,68,69].

Further insights into the function of the N-terminal domain of *B. subtilis* RNase Y will be obtained by studying its molecular interactions with its protein partners of the degradosome complex [28].

## 5. Conclusions

A coiled-coil is a structural element that is remarkable with respect to the diversity of conformations that it can adopt and for the range of functions that it exhibits [56]. As a result, long coiled-coil proteins encode an enormous repertoire of surface epitopes, in addition to potentially linking functional domains or communicating conformational changes [70]. There are several examples of proteins that have coiled-coils that act as a scaffold for interaction with other proteins or other domains. For example, the flexibility of the coiled-coil was shown to regulate the function of soluble guanylate cyclase [71,72].

Elongated structures, such as coiled-coils that are under-represented in the PDB or structures that have never been observed previously, are difficult to predict using AlphaFold. Indeed, it was previously reported that, in the CENP-E kinesin AlphaFold model, the structure of the motor domain was well predicted, whereas the flexible coiled-coil appeared folded like a ball, not representing a biologically and functionally relevant state [73]. In this example, the AlphaFold prediction, based on the monomer, did not resolve the coiled-coil structure, the fold of which depended on dimerization. Similarly, it was reported that the single chain-based predictions of coiled-coils-containing centriolar or centrosomal proteins, as well as the models of heterodimeric or multimeric coiled-coil assemblies, lacked structural plausibility [74]. Yet, AlphaFold has been shown to be successful in predicting long coiled-coils in several cases, such as human alpha or beta soluble guanylate cyclase [72].

Our multidimensional heteronuclear NMR study of Nter-BsRNaseY showed that this domain adopts a helix-type secondary structure over almost the entire sequence, in full agreement with the model calculated by AlphaFold. In our case, AlphaFold was also successful in predicting two conformations for the C-terminal helix, which were confirmed by our relaxation experiments. Although transient interactions are not expected to be captured by AlphaFold and the prediction of multimers is still at its beginning [75], here, we showed a case where AlphaFold was particularly useful to produce a reliable 3D model that helped to interpret the NMR data.

## Figures and Tables

**Figure 1 biomolecules-12-01798-f001:**
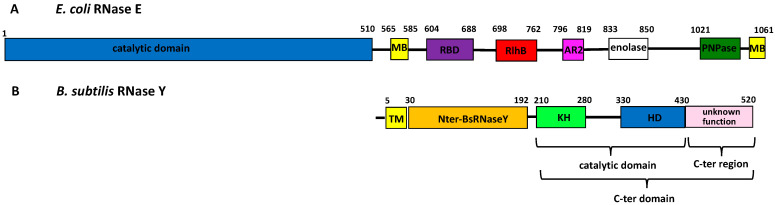
Comparison of the domain organization of *E. coli* RNase E and *B. subtilis* RNase Y. (**A**) Schematic representation of *E. coli* RNase E showing the interactions with proteins from the degradosome machinery (adapted from [19]). RBD and AR2 are arginine-rich RNA-binding domains flanking the RhlB (helicase) binding site and MB are membrane-binding domains. (**B**) Schematic representation of *B. subtilis* RNase Y. The catalytic domain contains KH and HD motifs. The KH homology module is a widespread RNA-binding motif, whereas the HD motif is characteristic of a superfamily of metal-dependent phosphohydrolases. TM is the transmembrane sequence. In this figure, the delimitation of the domains is based on the amino acids sequence.

**Figure 2 biomolecules-12-01798-f002:**
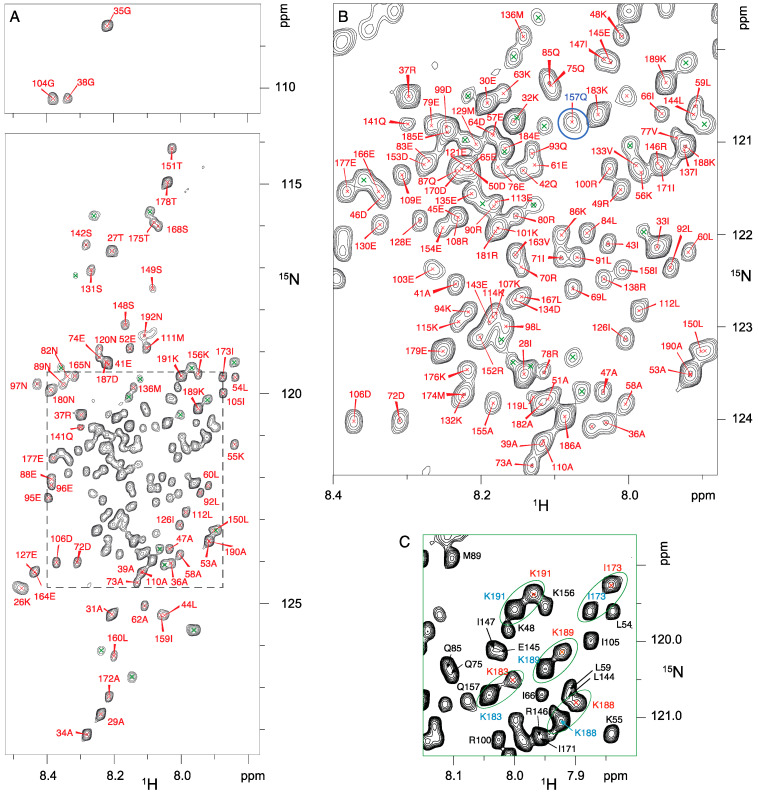
^1^H-^15^N BEST-TROSY spectrum of 970 µM ^15^N-^13^C-^2^D-labeled Nter-BsRNaseY. The spectrum was recorded at 950 MHz, in 40 mM MES buffer pH 6.8, 200 mM NaCl, 300K. (**A**) Overall spectrum and (**B**) enlarged view of framed region. Several peaks, such as Q157 (circled in blue), display a weak intensity. (**C**) Region of the ^1^H-^15^N BEST-TROSY spectrum of Nter-BsRNaseY showing several duplicated peaks, marked in blue (higher intensity peaks) and in red (lower intensity peaks). Pairs of split peaks are circled in green. The peaks that are not duplicated are marked in black.

**Figure 3 biomolecules-12-01798-f003:**
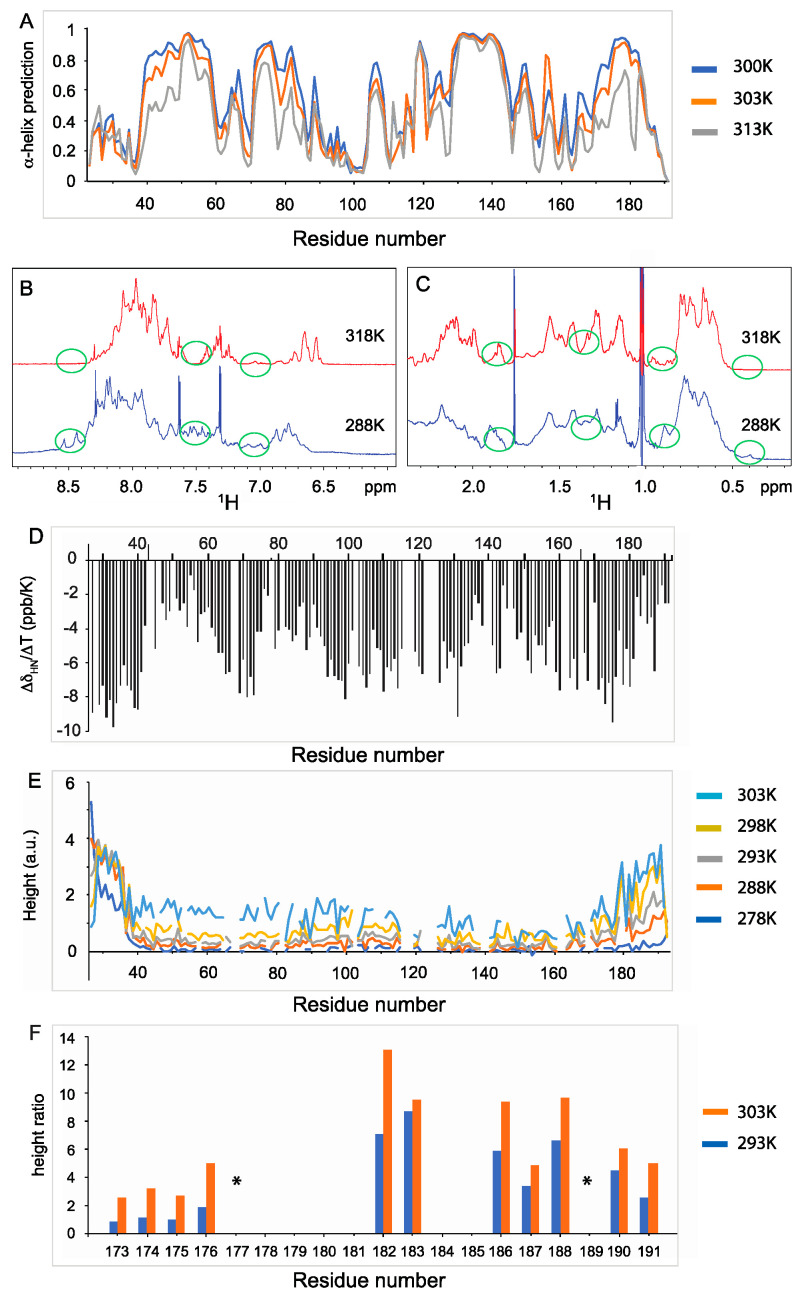
Flexibility and helix propensity of Nter-BsRNaseY amino acids, as deduced from the ^1^H-^15^N-^13^C NMR data. (**A**) Prediction of α-helices propensity by TALOS, derived from the backbone NMR chemical shifts at 300K (in blue), 303K (in orange), and 313K (in grey). Comparison of the amide and aromatic (**B**) and aliphatic protons (**C**) in the 1D NMR spectra recorded at 288K (in blue) and 318K (in red). The green circles highlight the highest differences observed in the chemical shifts at the two temperatures. (**D**) Temperature coefficients (∆δHN/∆T (ppb/K)). (**E**) ^1^H-^15^N TROSY peak heights (arbitrary units) at 278, 288, 293, 298, and 303K. The peak heights for residues 170 to 192 are the sum of the corresponding duplicated peaks in the ^1^H-^15^N TROSY spectra. (**F**) Comparison of the peak height ratio in the ^1^H-^15^N BEST-TROSY spectra of pairs of duplicated peaks for several residues located in the 173–191 segment at 293K (in blue) and 303K (in orange). The peaks in Appendix A were used to calculate the peak height ratios. * indicates that the peak height ratio could not be measured because the duplicated peaks of E177 and K189 are superimposed onto other peaks.

**Figure 4 biomolecules-12-01798-f004:**
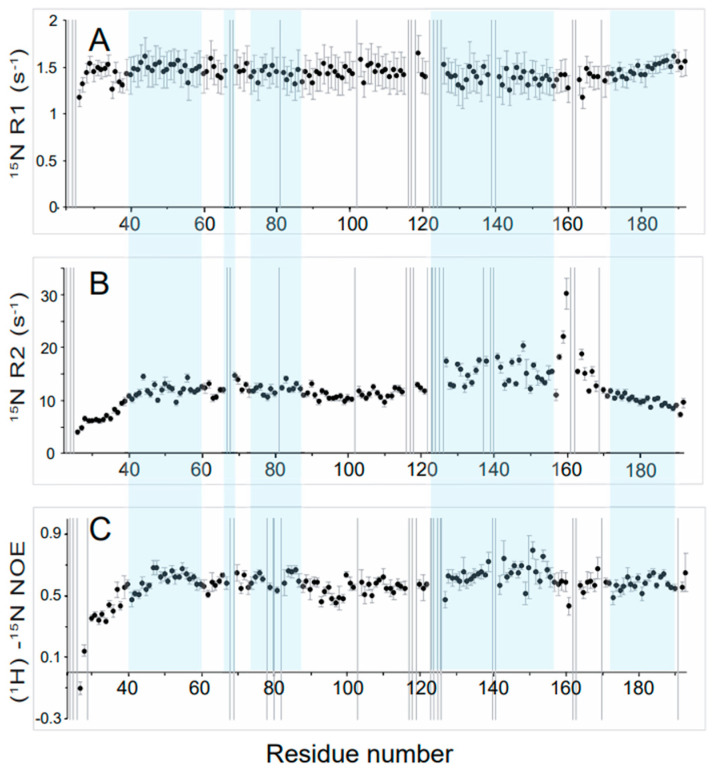
Relaxation rates and backbone dynamics of Nter-BsRNaseY. Plots of the ^15^N longitudinal relaxation rate R_1_ (**A**), ^15^N transverse relaxation R_2_ (**B**), heteronuclear {^1^H}-^15^N NOE, and (**C**) parameters obtained at 950 MHz ^1^H and 303K as a function of residue number. The unassigned residues of Nter-BsRNaseY are represented by vertical grey lines. The helices predicted by TALOS-N are highlighted in light blue.

**Figure 5 biomolecules-12-01798-f005:**
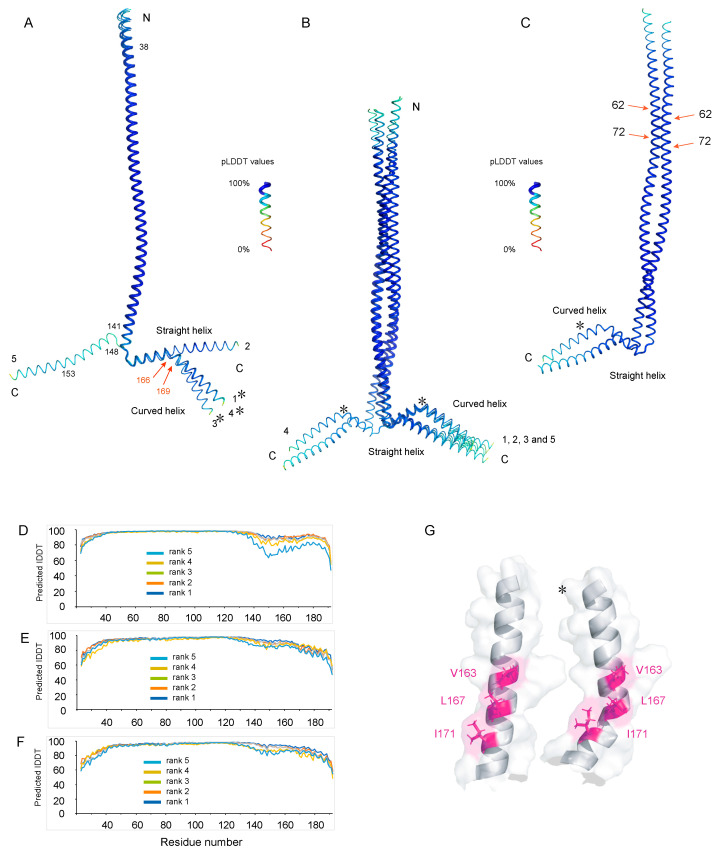
The five best 3D models of Nter-BsRNaseY generated by AlphaFold. The five models, numbered from 1 (best) to 5 (worst), were superimposed on the backbone of residues 33–138 that were shown to possess the best pLDDT values. (**A**,**B**): Superimposition of the AlphaFold models for the monomer (**A**) and dimer (**B**). (**C**) The best AlphaFold model of the dimer. The red arrows highlight the break in the supercoil. (**D**–**F**) pLDDT values for the AlphaFold models of the monomer (**D**) and the dimer: (**E**) first monomer and (**F**) second monomer. The residues were colored according to their pLDDT values, from red (0%) to blue (100%). (**G**) Hydrophobic interactions involving V163, L167, and I171, leading to the formation of a kink in the helical C-terminal extremity of the dimer helix. The curved helix is indicated by an asterisk.

**Figure 6 biomolecules-12-01798-f006:**
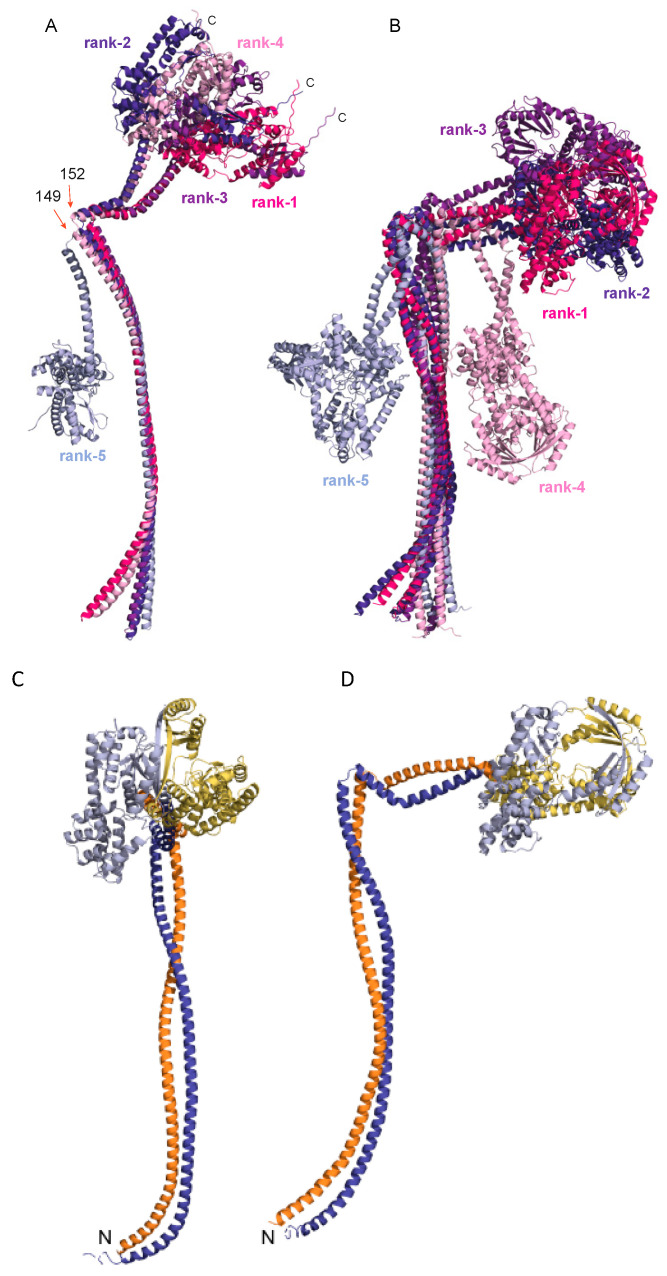
AlphaFold models of the full-length *B. subtilis* RNase Y. (**A**) The five best 3D models of the monomer. (**B**) The five best 3D models of the dimer. The models are numbered from 1 (the best) to 5 (the worst). The residues at the edges of the turn are highlighted as orange arrows. (**C**,**D**): Two different views showing the relative orientation of the two chains in the dimer of the best AlphaFold model for the full-length *B. subtilis* RNase Y. One chain is colored orange and yellow and the other is colored dark blue and light blue, for the N- and C-terminal domains, respectively.

## Data Availability

Resonance assignments for Nter-BsRNAseY at 303K were deposited in BMRB accession number 51229.

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
