# Peer review of "Structural Insights into the Dimeric Form of Bacillus subtilis RNase Y Using NMR and AlphaFold"

_biomolecules, 2022, doi:10.3390/biom12121798_

Round 1

Reviewer 1 Report

The authors provide a structural insight into the dimeric form of Bacillus Subtilis RNase Y  combining their NMR results with AlphaFold predictions. The NMR characterization of the dimer is well described and detailed. They show a great example of using AlphaFold to produce a 3D model that helped to interpret the NMR experimental data. The results and discussions are clear and solid. This reviewer recommend to accept this manuscript for publication in Biomolecules.

Author Response

Reviewer 1 did not ask for modifications.

Reviewer 3 Report

The authors previously expressed and characterized the secondary structural contents, monomer-dimer equilibrium and the molecular size of the N-terminal region of Bacillus subtilis RNase Y by using CD, SEC-MALS and SAXS, and concluded that their observations were consistent to the predicted coiled-coil form (ref 32). In the current study, the authors further characterized the N-terminal region using NMR and modeled the structure via alphaFold. The authors also did AphaFold calculation of the full-length protein, and demonstrated that the similar dimer formation at the N-terminal region was seen in the full length protein, too. Overall, as commented below, there are incorrect or speculative or qualitative descriptions, which make the manuscript difficult to read. 

Four major issues. 

(1) Essential data, such as dimer dissociation constant at different temperatures and protein concentration-dependence of the NMR spectra, are missing.  In section 3.1, the authors have to describe the dimer dissociation constants at 300 – 313 K first, to clarify what states were detected by NMR. Note that even a small fraction of one state may non-linearly affect signal intensity and/or chemical shifts of the other state. In addition, the authors have to show NMR spectra of the protein at different protein concentrations, to clarify whether the doubling of resonances depends on the protein concentration or not in section 3.2.  In figure 2 caption, the spectra were recorded at 0.97 mM protein concentration at 950 MHz. The authors should be able to decrease the concentration 50 fold at least. Normally, such a spectral feature, i.e., doubling of resonances, is described at the beginning of the NMR results, than at the end of section 3.2, to indicate what NMR experiments are needed.

(2) In section 3.1, the authors calculated S2 from TALOS-N (Figure 3F). When the monomer-dimer equilibrium affects chemical shifts, the S2 calculated from the shifts may be affected by the monomer-dimer equilibrium. Similarly, it is possible that the fold-unfold equilibrium may change the shifts and thus changes S2 as increase in the temperature. There is no description about the melting temperature and estimation of the fraction of the unfolded component. Although it is easy to calculate S2 from TALOS-N, it is not informative in such a complex system. Indeed, the intensity plot (Figure 3E) is rather similar to that of hetNOE (Figure 4C), but not to TALOS-N S2 (Figure 3F). I recommend the authors to remove Figure 3F, unless quantitative analysis of the S2 is newly added. 

(3) In section 4.1, the authors state “our NMR data cannot distinguish between parallel and anti-parallel coiled-coils” (line 564). There are several methods to identify it, including simple NMR PRE with N-terminal or C-terminal site, or cross linking or checking dimerization of differently truncated peptides. Alternatively, NOESY between a 13C/15N-labeled subunit and the unlabeled subunit can identify the binding site. EPR DEER experiments or fluorescence-labeling will also identify it. Rather than having a long discussion from line 564 to 581, the authors should do the experiments to conclude it. Otherwise, the authors have to remove the long discussion because the parallel form was expected anyway.

(4) Significance of the AlphaFold of the full-length protein (Section 3.5 and 4.2) is small. Since many ribonucleases are stabilized by metal ions, a simple computational fold of the catalytic domain may not be informative. Unless showing experimental results of the C-terminal region for dimerization, the authors have to shorten Section 3.5 and 4.2 and only focus on comparison of the structure of the N-terminal domain in full length with that of the N-terminal domain only. 

Minor points.

(5) I recommend the authors to remove some speculative descriptions from the Results section.  For example, “At pH 6.8, these histidines are probably involved in protonation/deprotonation processes”(line 249); “These results suggest that, at low temperature, the last C-terminal helix is involved in dimer formation and that this part of the dimer is the most affected by an increase in temperature, “ (line 300); “suggesting an easier disruption of the dimeric interactions for the former residues.”(line 329). 

(6) In Figure 3E, it is unclear whether the authors really plotted intensity of the cross peaks or the peak height. It is also unclear how the spectra recorded at different temperatures were normalized.  

(7) In section 3.3, the authors wrote “we also observed heterogeneity in the line widths of the 332 peaks in the 1H-15N BEST-TROSY” (line 332). The authors should change the word “heterogeneity” to “difference” or “variations”, because what they indicate is not discrete or severe different species. “Heterogeneity in the line widths” sounds like there are different species within the peak, making the sum of the linewidths to a shape that is not described by a single Lorentzian shape. 

(8) In section 3.3, the authors wrote “Moreover, low hetNOE and high 15N R2 values were also observed for the 158-170 segment (Figures 4B and 4C), probably resulting from a contribution to us-ms conformational or chemical exchange, as this segment has a low propensity to form helices (Figure 3A).”. This description is wrong. The authors have to separate the us-ms motion v.s. sub-nano second motion separately. NOE does not reflect us-ms motion effect because this time scale is slower than the molecular tumbling.

(9) In section 3.4, the authors state “The AlphaFold prediction is highly accurate for residues 38 to 169, with a predicted local-distance difference test value (pLDDT) > 90”. The word that the authors should use is not “accurate” but “converged”. High pLDDT (confidence) does not necessarily indicate high accuracy of the structure. If the protein or the protein region is disordered, pLDDT may not become high.

(10)  Line 145 “2.2.3. D-structure” to “2.2. 3D-structure”.

Round 2

Reviewer 3 Report

The authors were responsible for the revision.